# Ground state solutions of inhomogeneous Bethe equations

**Samuel Belliard[1,2]\* and Alexandre Faribault[3]**

**1** Sorbonne Université, CNRS, Laboratoire de Physique Théorique et Hautes Energies,
LPTHE, F-75005, Paris, France
**2** Institut de Physique Théorique, DSM, CEA, URA2306 CNRS Saclay, F-91191, Gif-sur-Yvette,
France
**3** Université de Lorraine, CNRS, LPCT, F-54000 Nancy, France

⋆ samuel.belliard@gmail.com

## Abstract

The distribution of Bethe roots, solution of the inhomogeneous Bethe equations, which characterize the ground state of the periodic XXX Heisenberg spin-$\frac{1}{2}$ chain is investigated. Numerical calculations show that, for this state, the new inhomogeneous term does not contribute to the Baxter T-Q equation in the thermodynamic limit. Different families of Bethe roots are identified and their large N behaviour are conjectured and validated.

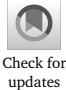
## 1 Introduction

The Bethe-Hulthén ansatz approach [1, 2] has shown its interest by allowing on to investigate the thermodynamic limit of spin chains [2, 4–9] and predict exact physical quantities. The spectrum of the Hamiltonians of integrable spin chains follows from the one of the transfer

matrix $t(\lambda)$, the generating function of conserved quantities of the model and can be characterized by the Baxter T-Q equation [11] which, in the case of the periodic XXX Heisenberg spin$-\frac{1}{2}$ chain with $N$ spins is given by

$$t(\lambda)\hat{q}(\lambda) = \left(\lambda + \frac{i}{2}\right)^N \hat{q}(\lambda - i) + \left(\lambda - \frac{i}{2}\right)^N \hat{q}(\lambda + i), \tag{1}$$

where $\hat{q}(\lambda)$ is the Q operator which commutes with the transfer matrix and whose eigenvalues have the polynomial form

$$q(\lambda) = \prod_{k=1}^{M}(\lambda - \lambda_k), \tag{2}$$

with $M \leq \frac{N}{2}$.

The recent development of the Bethe ansatz for the models without $U(1)$ symmetry, where the usual Bethe-Hulthén ansatz approach fails, leads to the discovery of a new family of Baxter T-Q equation [16], called *inhomogeneous Baxter T-Q equation* or *modified Baxter T-Q equation*. This new family differs from the usual one in two ways: it involves a new term and the order of the associated Q polynomial is fixed. The implementation of the Bethe ansatz changes and the techniques to obtain this equation and the associated states lies at the intersection of many different methods: Off diagonal Bethe ansatz [15], Modified algebraic Bethe ansatz [12, 24], Separation of variables [20–22]. Moreover, it was observed in [15] that this *inhomogeneous Baxter T-Q equation* can also characterize finite models with $U(1)$ symmetry and, in the case of the periodic XXX Heisenberg spin$-\frac{1}{2}$ chain with $N$ spins, is given by [1]

$$t(\lambda)\hat{Q}(\lambda) = -\left(\lambda + \frac{i}{2}\right)^N \hat{Q}(\lambda - i) - \left(\lambda - \frac{i}{2}\right)^N \hat{Q}(\lambda + i) + 4\left(\lambda^2 + \frac{1}{4}\right)^N, \tag{3}$$

where $\hat{Q}(\lambda)$ is the inhomogeneous Q operator whose eigenvalues have the polynomial form

$$Q(\lambda) = \prod_{i=1}^{N}(\lambda - u_i). \tag{4}$$

It is worth mentioning that the existence of such an operatorial T-Q equation is a conjecture contrary to its functional form.

The understanding of the thermodynamic limit for models with a spectrum characterized by such modified Baxter T-Q equation is an important challenge. For models with a boundary term, it was observed numerically in [15, 17–19] that the contribution to the ground state energy for the inhomogeneous term decreases as $N^{-1}$, where $N$ is the length of the chain.

In this work, we investigate the evolution of the ground state's Bethe roots as a function of the chain length $N$ and classify the types of roots which appear in the solution of the inhomogeneous Bethe equations. In order to perform the thermodynamic limit a key ingredient is, indeed, knowledge of the structure of the Bethe equation's solution. For example, the periodic XXX Heisenberg spin$-\frac{1}{2}$ chain, the antiferromagnetic ground state belongs to the class of solution where $q(\lambda)$ have only real roots [2–5] (while excited states can also have complex roots that behave as strings [1]). Considering that the Hamiltonian of this model can be characterized by both the homogeneous Baxter T-Q equation (1) and the inhomogeneous Baxter T-Q equation (3), one can retrieve, using the well-know homogeneous case, the Bethe roots of

---

[1]A more general characterization depending on an arbitrary parameter can be constructed [16], here for numerical convenience we fix it to its simplest form which also corresponds to the diagonal limit of the T-Q equation for the twisted XXX Heisenberg spin$-\frac{1}{2}$ chain in the parametrization of [24].

the inhomogeneous Baxter Q polynomial, without directly solving the inhomogeneous Bethe equations. Here, we will exclusively consider the distribution of Bethe roots which characterizes the antiferromagnetic ground state.

The paper is organized as follows, in the section 2 we recall the Bethe equations, their logarithmic form, and their relation to the homogeneous Baxter T-Q equation (1). The ground state's real Bethe roots are then calculated up to $N = 300$. In section 3 we reconstruct the Q polynomial from the inhomogeneous Baxter T-Q equation (3) and the homogeneous solution giving the transfer matrix in order to numerically find the solution of the inhomogeneous Bethe equations characterizing the antiferromagnetic ground state of the model. Three different families of Bethe roots are then identified and the large $N$ behavior is discussed. Concluding remarks are given in section 4.

## 2 Homogeneous ground state solution

The roots of the Q baxter polynomial $q(\lambda)$ (2), $\{\lambda_1 \ldots \lambda_M\}$ satisfy the Bethe equations

$$\left(\lambda_k + \frac{i}{2}\right)^N \prod_{j \neq k}^M (\lambda_k - \lambda_j - i) = \left(\lambda_k - \frac{i}{2}\right)^N \prod_{j \neq k}^M (\lambda_k - \lambda_j + i), \tag{5}$$

with $k \in \{1, ..., M\}$ and $M = \{0, 1, ..., N/2\}$ (we always consider $N$ even). These equations follow from the Baxter TQ equation (1) evaluated at the zeros of $q(\lambda)$. The knowledge of the Bethe roots $\{\lambda_1 \ldots \lambda_M\}$ allows from equation (1) a direct numerical computation of the eigenvalue $t(\lambda)$ at any point $\lambda$. This formulation would lead to numerical difficulties in evaluating $t(\lambda)$ in the immediate vicinity of a Bethe root, as we should divide by $q(\lambda)$, a problem which is explicitly avoided by the numerical approach used in the following. Instead of trying to directly solve the Bethe equations in the form (5), it is much more stable for numerical computation to use the logarithmic form

$$N \ln \left( \frac{1 - 2i\lambda_j}{1 + 2i\lambda_j} \right) = i \left( N + M + 1 + J_j \right) \pi + \sum_{k \neq j}^M \left[ \ln \left( \frac{1 - i \left( \lambda_j - \lambda_k \right)}{1 + i \left( \lambda_j - \lambda_k \right)} \right) \right], \tag{6}$$

where $J_j$ is an integer which specifies the various possible logarithmic branches. For a size, $N = 4n$, such that both $N$ and $M = N/2$ are even, the ground state solution is found for the values $J_j = 2(j - 1 - N)$ with $j = 1, 2 \ldots, \frac{N}{2}$.

A simple implementation of the Newton-Raphson method is then sufficient for equation (6) to rapidly converge to the desired ground state solution starting from almost any initial approximation for the position of the Bethe roots. One can exploit the symmetry of the ground state root distribution which come in pairs of $\pm\lambda_i$, in order to reduce the size of the system by half. Indeed, one can solve for only the positive roots: $\lambda_1 \ldots \lambda_{N/4}$ by rewriting the $N/4$ first equations in terms of these variables having replaced the remaining ones by $\{\lambda_{N/4+1} \ldots \lambda_{N/2}\} = \{-\lambda_1 \cdots -\lambda_{N/4}\}$. The resulting solutions are presented in Figure 1 and uniquely define the corresponding $q(\lambda)$ polynomial, i.e. the $N/2$ order polynomial with zeros placed at each of these roots.

## 3 Inhomogeneous T-Q ground state solution

The roots of the Q baxter polynomial $Q(\lambda)$ (4), $\{u_1 \ldots u_N\}$ satisfy the Bethe equations

$$\left( u_k - \frac{i}{2} \right)^N \prod_{j \neq k}^N (u_k - u_j + i) - \left( u_k + \frac{i}{2} \right)^N \prod_{j \neq k}^N (u_k - u_j - i) = 4 \left( u_k^2 + \frac{1}{4} \right)^N, \tag{7}$$

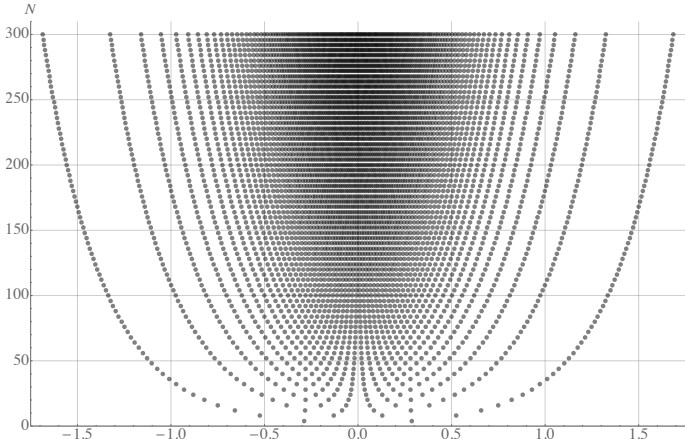

Figure 1: Position of the $N/2$ Bethe roots for the ground-state solution to the homogeneous T-Q equation. System sizes range from $N = 4$ to $N = 300$ with $N/2$ even.

with $k = \{1, .., N\}$. These equations follow from the Baxter TQ equation (3) evaluated at the zeros of $Q(\lambda)$.

Having computed the homogeneous $q$ polynomial through its $N/2$ zeros in the previous section and therefore the corresponding $t(\lambda)$ polynomial which stays the same in inhomogeneous case and the homogeneous one, we can retrieve the inhomogeneous Baxter Q operator from the inhomogeneous T-Q equation (3) and find it roots without having to explicitly solve the inhomogeneous Bethe equations (7). This greatly simplifies the calculation since, due to the inhomogeneous term, they cannot be simply brought into logarithmic form so that their explicit numerical solving is a much more arduous task than in the homogeneous case.

### 3.1 Reduction of the inhomogeneous T-Q equation

Considering that for any $\lambda$, equation (3) couples exclusively the polynomial $Q$ at three points $\lambda, \lambda \pm i$, it becomes simple to define a system of sparse linear equations. Indeed, through a Lagrange polynomials representation, one can define any polynomial $P(\lambda)$ of order $N$, in terms of the values $P(z_i)$ it takes at a set of $N + 1$ arbitrary grid points $\{z_1 \ldots z_{N+1}\}$. In the case at hand (4), since the leading coefficient is know to be 1, only $N$ points are necessary to define exactly

$$Q(\lambda) = \ell(\lambda) \left[ 1 + \sum_{n=1}^{N} \frac{Q(z_n)}{\ell'(z_n)} \frac{1}{\lambda - z_n} \right] \quad \text{with} \quad \ell(\lambda) \equiv \prod_{n=1}^{N} (\lambda - z_n). \tag{8}$$

Considering that the T-Q equation only couples $Q(\lambda)$ and $Q(\lambda \pm i)$, one can choose to use a grid of points $\{z_1 \ldots z_N\}$, which are separated by $i$ allowing us to fully define $Q(\lambda)$ in terms of the $N$ values $Q(z_i)$. These $Q(z_i)$ will simply be the solution to a set of (mostly) tridiagonal linear equations. Indeed, on a grid such that $z_{n+1} = z_n + i$ , at every point except for $z_1$ and $z_N$, equation (3) reads

$$t(z_n)Q(z_n) + \left( z_n + \frac{i}{2} \right)^N Q(z_{n-1}) + \left( z_n - \frac{i}{2} \right)^N Q(z_{n+1}) = 4 \left( z_n^2 + \frac{1}{4} \right)^N, \tag{9}$$

leading to a core set of $N - 2$ linear equations coupling only three of the variables $Q(z_n)$. The two extreme points $z_1$ and $z_N$ still lead to full equations with non-zero coefficients in front of each variables $\{Q(z_1), \ldots, Q(z_{N+1})\}$ due to the presence of $Q(z_1 - i)$ or $Q(z_{N+1} + i)$ which, using equation (8), could be expressed in terms of a full sum over the values $\{Q(z_1), \ldots, Q(z_N)\}$.

When considering ground state solutions for even $N/2$ ($N$ being an integer multiple of 4) we know, from the previous section, that the $t(\lambda)$ polynomial can be explicitly deduced from the homogeneous Baxter $T-Q$ equation with a $q(\lambda)$ polynomial whose zeros are all real and come in symmetric pairs: $\pm\lambda_i$. This fact allows us to conclude that

$$t(z) = t(-z) = t^*(z^*) \qquad \text{which implies} \qquad Q(z) = Q(-z) = [Q(z^*)]^*. \tag{10}$$

It therefore leads to further simplifications to choose the ($i$-separated) grid points $z_n$ to lie on the imaginary axis and to be symmetric around the real axis, namely: $z_n = \left(-\frac{N+1}{2} + n\right)i$ for $n = 1, 2 \ldots N$. By doing so, one insures that $Q(z_i) \in \mathbb{R}$ and that grid points come in pairs, i.e. there exists $z_n = -z_{n'}$ allowing us to reduce the system of linear equations to only $N/2$ equations using variables $Q(z_i)$ which will take real values.

Additionally, it is easily shown that, for any $N$ integer multiple of 4, a pair of Bethe roots is always found at $\pm\frac{i}{2}$. Indeed looking at the specific equations obtained from (3) at $\pm\frac{i}{2}$, one has:

$$t(i/2)Q(i/2) + Q(-i/2) = 0, \qquad t(-i/2)Q(-i/2) + Q(i/2) = 0. \tag{11}$$

Considering as well that $t(-i/2) = t(i/2)$ and $Q(i/2) = Q(-i/2)$, we simply find:

$$[t(i/2) + 1]Q(i/2) = 0, \tag{12}$$

and therefore to $Q(i/2) = Q(-i/2) = 0$ since equation (1) gives $t(i/2) = \frac{q(-i/2)}{q(i/2)} = 1$.

Having shown that $Q(z)$ has a pair of roots at $\pm\frac{i}{2}$ and that our choice of a purely imaginary symmetric set of grid points allows us to reduce the expression for $Q(z)$ by half, one can define:

$$Q(z) = \left(z - \frac{i}{2}\right)\left(z + \frac{i}{2}\right)\tilde{Q}(z), \tag{13}$$

where $\tilde{Q}(z)$ is a symmetric polynomial ($\tilde{Q}(z) = \tilde{Q}(-z)$) of order $N-2$ with leading coefficient 1 which obeys the $T-\tilde{Q}$ equation:

$$
\begin{aligned}
t(\lambda)\tilde{Q}(\lambda) \;=\; & -\left(\lambda + \frac{i}{2}\right)^{N-1}\left(\lambda - \frac{3i}{2}\right)\tilde{Q}(\lambda - i) - \left(\lambda - \frac{i}{2}\right)^{N-1}\left(\lambda + \frac{3i}{2}\right)\tilde{Q}(\lambda + i) \\
& + 4\left(\lambda^2 + \frac{1}{4}\right)^{N-1}.
\end{aligned}
\tag{14}
$$

The polynomial $\tilde{Q}(z)$ can be written, in a Lagrange form, only in terms of the grid points in the upper complex plane as:

$$\tilde{Q}(z) = \tilde{\ell}(z) \sum_{n=1}^{N/2-1} \frac{\tilde{Q}\left(\frac{2n+1}{2}i\right)}{\tilde{\ell}'\left(\frac{2n+1}{2}i\right)}\left[\frac{1}{z - \left(\frac{2n+1}{2}\right)i} - \frac{1}{z + \left(\frac{2n+1}{2}\right)i}\right],$$

$$\text{with } \tilde{\ell}(z) = \frac{\ell(z)}{(z - i/2)(z + i/2)} = \prod_{n=1}^{N/2-1}\left(z - \left(\frac{2n+1}{2}\right)i\right)\left(z + \left(\frac{2n+1}{2}\right)i\right), \tag{15}$$

where we used the antisymmetry relation: $\tilde{\ell}'\left(\frac{2n+1}{2}i\right) = -\tilde{\ell}'\left(-\frac{2n+1}{2}i\right)$.

In the end, the different values $\{\tilde{Q}(3i/2), \tilde{Q}(5i/2), \tilde{Q}(7i/2)\ldots\tilde{Q}((N-1)i/2)\}$ needed to reconstruct the polynomial $\tilde{Q}(z)$ (and therefore $Q(z)$) are found as the solution to the linear system of equations:

$$t(3i/2)\tilde{Q}(3i/2) + 3\tilde{Q}(5i/2) + 2^{N+1} = 0$$

$$
\begin{aligned}
& t(ki + i/2)\tilde{Q}(ki + i/2) + (k+1)^{N-1}(k-1)\tilde{Q}((k-1)i + i/2) \\
& \quad + (k+2)(k)^{N-1}\tilde{Q}((k+1)i + i/2) = -4(k(k+1))^{N-1} \quad \forall\, k = 2, 3 \ldots N/2 - 2
\end{aligned}
$$

$$t(0)2^N\tilde{Q}(0) - 6\tilde{Q}(i) - 2^{4-N} = 0. \tag{16}$$

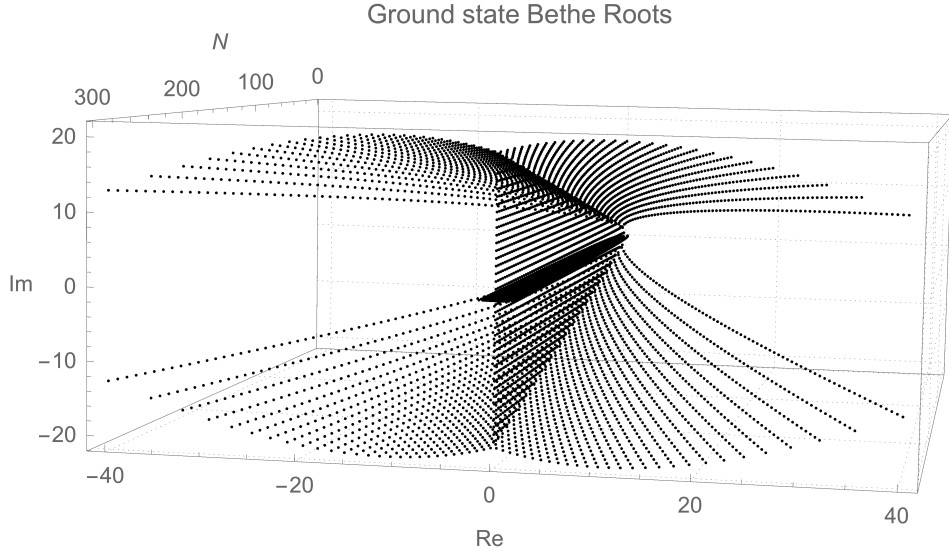

Figure 2: Inhomogeneous Bethe roots for N=4 to N=300.

The last equation, completing the system, is simply obtained from (14) evaluated at $\lambda = 0$. It is the only one of these equations which involves every $\tilde{Q}(z_i)$ when explicitly written in terms of $\{\tilde{Q}(3i/2)\ldots\tilde{Q}((N-1)i/2)\}$ using the Lagrange basis representation (15) of $\tilde{Q}(0)$ and $\tilde{Q}(i)$. While the resulting linear system requires more than machine precision to be solved numerically even for modest chain lengths $N$, its sparsity allows one to go to higher precision calculations in what remains a reasonable computation time, even on a standard personal computer.

Having then found the set of $\{\tilde{Q}(3i/2),\tilde{Q}(5i/2),\tilde{Q}(7i/2)\ldots\tilde{Q}((N-1)i/2)\}$, one can use any standard solver, at high precision, to find the roots of the resulting $\tilde{Q}$ polynomial, by simply numerically finding the points at which equation (15) equals 0.

## 3.2 Numerical solution and roots families

The general structure of the ground state solution obtained numerically, see figure 2, is three-fold. First, it systematically contains a series of $N/2$ real roots. Secondly, there is a string of purely imaginary roots and finally a series of roots with both non-zero real and imaginary parts which create four symmetric arcs which emerge from both ends of the imaginary string.

The general result is best visualised when plotting each of these substructures individually, and therefore, the real-valued roots are first plotted in figure 3.

Beyond the realization that the number of such real roots is indeed systematically the same as the number of roots in the homogeneous case, namely $N/2$, one easily infers from the plot, that, in the large $N$ limit, this set of real roots will correspond exactly to the homogeneous solution. It will therefore become dense in the thermodynamic limit.

In figure 4 we exclusively plot the purely imaginary roots as well as the number of such roots in figure 5 for system sizes $N$ which are integer multiples of four.

As can be seen from large enough system sizes, these imaginary roots tend to arrange into of perfect symmetric string of $i$-separated Bethe roots. The only visible deviations from this ideal string structure appear at the very top and bottom ends of the string.

Within the studied range, the number $N_I$ of such imaginary roots stays bounded by $\frac{N}{8} \leq N_I \leq \frac{N}{8} + \frac{9}{2}$. The upper bound corresponds exactly to $N_I$ for $N = 44, N = 76, N = 108$ and $N = 140$, while the lower bound does so for $N = 256$ and $N = 288$. While this does not

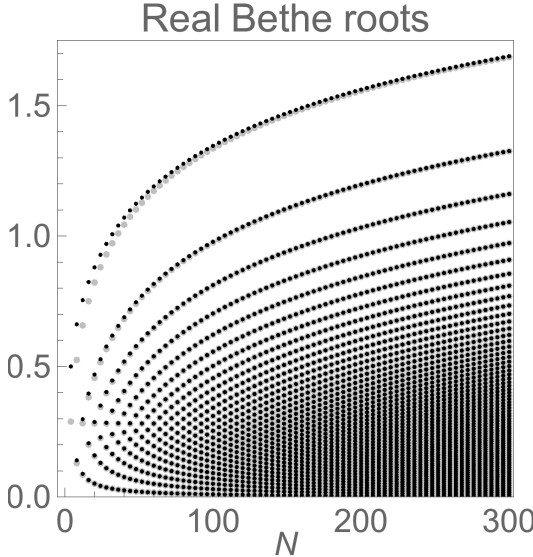

Figure 3: Positive real Bethe roots solution to the inhomogeneous T-Q equation (black dots) and positive Bethe roots solution of the homogeneous T-Q equation (gray circles).

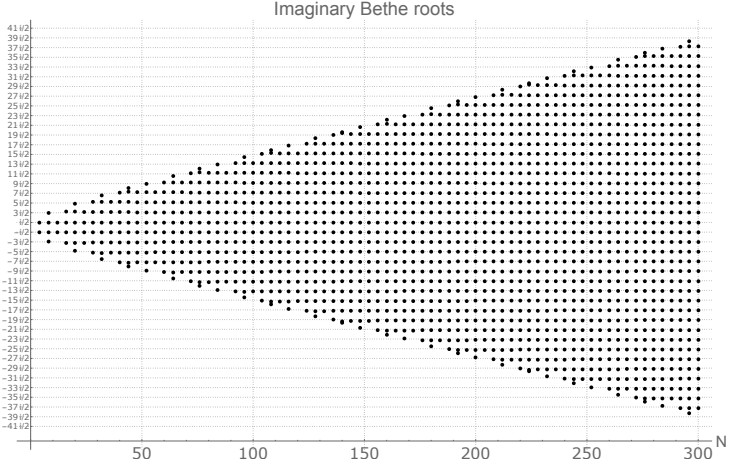

Figure 4: Imaginary Bethe roots for system sizes N=4 to N=300.

allow one to conjecture that these bound hold true for arbitrary system sizes, it seems safe to conjecture that, as system size grows, the length of this imaginary string will also keep growing. Indeed, although when going from $N$ to $N + 4$, it is possible to see $N_I$ go down (by 2) therefore giving a shorter string, comparing any $N$ to the corresponding $N + 8$ it is found that $N_I$ is never reduced.

The last subset of roots for which both real and imaginary parts take non-zero values form the arc-like structures and are explicitly shown in figure 6 for the upper right complex plane.

Higher values of $N$ always lead to arcs whose both end points are further from the real axis, so that, in the figure, they naturally appear as ordered by chain length N. Contrarily to the real roots which densify as $N$ becomes large, the roots on these arcs actually get further apart from one another as the system size gets larger. Consequently, one cannot expect, in the thermodynamic limit $N \to \infty$ to be in a position to define a continuous root density to properly describe these structures.

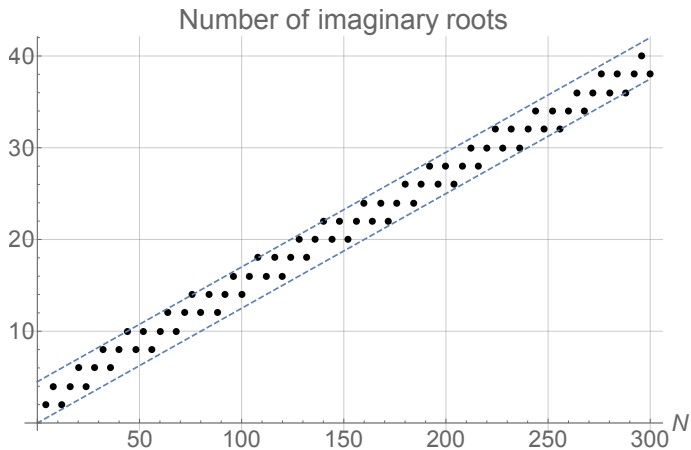

Figure 5: Number of imaginary Bethe roots and the bounds $\frac{N}{8}$ and $\frac{N}{8} + \frac{9}{2}$

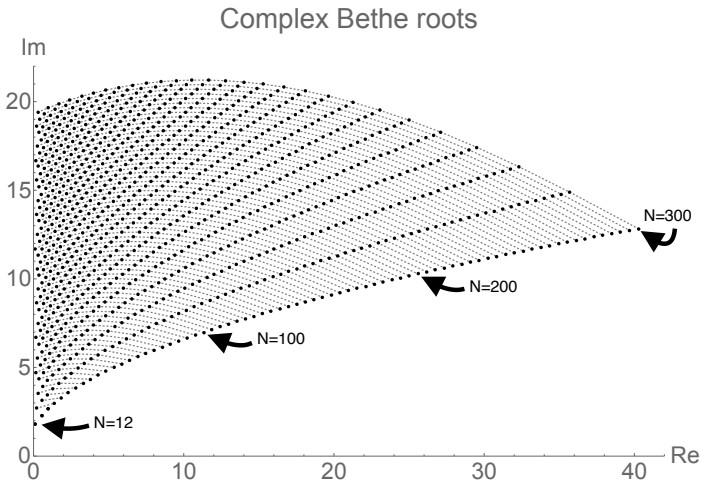

Figure 6: Complex Bethe roots in the upper right complex plane. Each arc (dashed lines) corresponds to a given N, ranging from $N = 12$ (lowest point) to $N = 300$ (highest arc). Three symmetric arcs also exist in the three other quadrants of the complex plane.

Nonetheless, these numerical results support strongly the conjecture that, as $N \to \infty$, these complex roots will all be such that their norm is also going to infinity.

The conjectures made from these finite size results can now be explicitly verified to give the correct solution to the inhomogeneous T-Q equation in the thermodynamic limit $N \to \infty$. Indeed, we have this threefold structure for which:

1- $N/2$ real Bethe roots $u_a^r$ will tend to the solution of the homogeneous problem.

2- $N_I$ imaginary Bethe roots $u_a^i$ that will form an infinitely long string of $i$-separated roots, with $N_I$ going to infinity.

3- The remaining $N/2 - N_I$ roots $u_a^c$ form discrete points on an arc infinitely far from the dense distribution of real roots $|u_a^c| >> |u_b^r|$.

Under these assumptions, the Q polynomial can be decomposed in three parts

$$Q(\lambda) = q^R(\lambda) q^I(\lambda) q^C(\lambda) \tag{17}$$

with

$$q^R(\lambda) = \prod_{a=1}^{N/2}(\lambda - u_a^r), \quad q^I(\lambda) = \prod_{a=1}^{N_I}(\lambda - u_a^i), \quad q^C(\lambda) = \prod_{a=1}^{N/2-N_I}(\lambda - u_a^c) \tag{18}$$

which allows one to rewrite the inhomogeneous T-Q equation (3) as

$$t(\lambda)q^R(\lambda) = -\left(\lambda + \frac{i}{2}\right)^N q^R(\lambda - i)\frac{q^I(\lambda - i)q^C(\lambda - i)}{q^I(\lambda)q^C(\lambda)} \tag{19}$$

$$-\left(\lambda - \frac{i}{2}\right)^N q^R(\lambda + i)\frac{q^I(\lambda + i)q^C(\lambda + i)}{q^I(\lambda)q^C(\lambda)} + 4\frac{\left(\lambda^2 + \frac{1}{4}\right)^N}{q^I(\lambda)q^C(\lambda)}. \tag{20}$$

The roots belonging to the complex arcs are such that $|u_b^c| \to \infty$ when $N \to \infty$ and therefore, in the same limit we have

$$\frac{q^C(\lambda \pm i)}{q^C(\lambda)} \to 1. \tag{21}$$

For arbitrary values of $\lambda$ such that $|\lambda| << |u_a^c|$ for every $a = 1, 2 \ldots N/2 - N_I$, the inhomogeneous term becomes negligible compared to the three other ones and therefore, in this whole small $|\lambda|$ region:

$$\frac{\left(\lambda^2 + \frac{1}{4}\right)^N}{q^I(\lambda)q^C(\lambda)} \to 0. \tag{22}$$

On the other hand, the contribution from the string of imaginary roots can be simplified when $N_I$ goes to infinity. Indeed for the pure $i$-separated string, we have

$$\lambda_b^I = -iN_I/2 + (b-1)i \tag{23}$$

and thus

$$q^I(\lambda - i) = \prod_{b=1}^{N_I}(\lambda - i - \lambda_b^I) = \prod_{b=1}^{N_I}(\lambda + iN_I/2 - (b-2)i) = q^I(\lambda)\frac{\lambda - iN_I/2 + i}{\lambda + iN_I/2 + i} \tag{24}$$

$$q^I(\lambda + i) = \prod_{b=1}^{N_I}(\lambda + i - \lambda_b^I) = \prod_{b=1}^{N_I}(\lambda + iN_I/2 - b i) = q^I(\lambda)\frac{\lambda + iN_I/2 - i}{\lambda - iN_I/2 - i} \tag{25}$$

It follows that, for a string length $N_I \to \infty$, one has:

$$\frac{q^I(\lambda \pm i)}{q^I(\lambda)} \to -1. \tag{26}$$

Combining those two limits of $q^I$ and $q^C$ for $N \to \infty$ we finally fall back (for small $|\lambda|$), on the homogeneous Baxter T-Q equation:

$$t(\lambda)q^R(\lambda) = \left(\lambda + \frac{i}{2}\right)^N q^R(\lambda - i) + \left(\lambda - \frac{i}{2}\right)^N q^R(\lambda + i). \tag{27}$$

Since a polynomial equation which is verified for a whole sector (small $|\lambda|$) is, by extension, verified everywhere, the remaining roots $u_a^r$ are then indeed the solution of the homogeneous T-Q equation. The large $N$ limit conjectures made above therefore provide, in the thermodynamic limit, the solution to the inhomogeneous problem. Therefore, for the ground state, the inhomogeneous term does not contribute to the Baxter T-Q equation in the thermodynamic limit.

## 4  Conclusion

In this paper we investigated the distribution of the Bethe roots which are solution to the inhomogeneous Bethe equation and characterise the antiferromagnetic ground state of the Heisenberg XXX spin-$\frac{1}{2}$ chain. We observe that for this state, under some simple conjectures about the behavior of the Bethe roots as function of $N$, the inhomogeneous Baxter T-Q equation reduces to the homogeneous one. This results from two facts: firstly a large set of complex roots go to infinity as $N \to \infty$ while secondly an infinite discrete purely imaginary string of roots changes the sign in front of the two first terms of the inhomogeneous Baxter T-Q equation.

As already mentioned, the inhomogeneous Baxter T-Q equation (3) is not the unique parametrization (see footnote p 2). In the general case we have an arbitrary parameter $\alpha$ and the inhomogeneous Baxter T-Q equation reads

$$t(\lambda)Q(\lambda) = \alpha \left( \lambda + \frac{i}{2} \right)^N Q(\lambda - i) + \alpha^{-1} \left( \lambda - \frac{i}{2} \right)^N Q(\lambda + i) + (2 - \alpha - \alpha^{-1}) \left( \lambda^2 + \frac{1}{4} \right)^N . \quad (28)$$

In this case, it is much more complex to solve numerically for large $N$ due to the additional parameter and the asymmetric form of the equation. However, one could still expect a similar threefold structure to emerge, now characterized by:

$$\frac{q^I(\lambda \pm i)}{q^I(\lambda)} \to \alpha^{\pm 1}.$$

We can also extend this hypothesis to the case of the XXX spin chain with an arbitrary twist (as considered in [24]). In that case one also has both homogeneous and inhomogeneous TQ equations. Namely:

$$t(\lambda)q(\lambda) = \kappa_1 \left( \lambda + \frac{i}{2} \right)^N q(\lambda - i) + \kappa_2 \left( \lambda - \frac{i}{2} \right)^N q(\lambda + i), \quad (29)$$

where the two parameters $\kappa_i$ satisfy $\kappa_1 + \kappa_2 = \kappa + \tilde{\kappa}$, $\kappa_1 \kappa_2 = \kappa \tilde{\kappa} - \kappa^+ \kappa^-$, and

$$t(\lambda)Q(\lambda) = (\tilde{\kappa} - \rho) \left( \lambda + \frac{i}{2} \right)^N Q(\lambda - i) + (\kappa - \rho) \left( \lambda - \frac{i}{2} \right)^N Q(\lambda + i) + 2\rho \left( \lambda^2 + \frac{1}{4} \right)^N , \quad (30)$$

with $\rho^2 - \rho(\kappa + \tilde{\kappa}) + \kappa^+ \kappa^- = 0$. In that case one could still expect a similar threefold structure to emerge, now characterised by:

$$\frac{q^I(\lambda - i)}{q^I(\lambda)} \to \frac{\kappa_1}{\tilde{\kappa} - \rho}, \qquad \frac{q^I(\lambda + i)}{q^I(\lambda)} \to \frac{\kappa_2}{\kappa - \rho},$$

where we also have $\frac{\kappa_1}{\tilde{\kappa} - \rho} = \frac{\kappa - \rho}{\kappa_2}$.

It would be of strong interest to extend this numerical analysis to excited states of the periodic Heisenberg XXX spin-$\frac{1}{2}$ chain as well as for models where the $U(1)$ symmetry is broken (as the Heisenberg XXX spin-$\frac{1}{2}$ chain with boundaries). In the later case, the trick to use the homogeneous solutions would fail and one should therefore try to solve directly the inhomogeneous Baxter equation or use an explicit diagonalisation of the transfer matrix to retrieve the roots of the Baxter Q polynomial.

Having identified the Bethe roots distribution for the inhomogeneous Baxter TQ equation, we should be in a position to consider possible analytical results along the lines of Yang and Yang's works, and many others (see for recent development [23]). This question should be discussed elsewhere.

One of the important questions would be to see if it is possible or not to find a model (or a specific state) where the inhomogeneous term can contribute to the Baxter T-Q equation in the thermodynamic limit. Actually, the limited number of cases considered so far in the literature have all shown that it is not the case.

**Acknowledgements:** We thank V. Pasquier, B. Vallet, P. Baseilhac, N. Crampé, D. Serban, N. Slavnov, R. Pimenta, K.K. Kozlowski for discussions and their interest. S. B. would like to thank the LPCT of Université de Lorraine for its hospitality and the LMPT of Tours University for the invitation during which a part of this work was done. A. F. would also like to thank the LMPT of Tours University for the invitation during which parts of this research was carried out.

**Funding information:** S.B. was supported by a public grant as part of the Investissement d'avenir project reference ANR-11-LABX-0056-LMH, LabEx LMH.

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
