# Peer review of "Ground state solutions of inhomogeneous Bethe equations"

_SciPost Physics, doi:SciPost Phys. 4, 030 (2018)_

## Round 2 · Referee Report · Anonymous (Referee 1) · 2018-4-20

Strengths

1- First analysis of Bethe roots densities within the inhomogeneous TQ relations

2- An original method to study numerically the inhomogeneous TQ relations

Weaknesses

1- It is not clear whether one can generalize the method to perform the same analysis for models which don't have a presentation with homogeneous TQ relations

Report

The authors investigate numerically the density of Bethe roots of the Bethe equations for XXX model with periodic conditions (for the ground state) and presented with the inhomogeneous TQ relations.

Although this density is known (within the homogeneous TQ relations), the aim is to get some insight on the inhomogeneous TQ relations, which can be used to deal with open or twisted boundary conditions.
The inhomogeneous TQ relations are difficult to study directly numerically because the inhomogeneous term prevents (a priori) to use the logarithmic version of the equations (contrarily to the homogeneous case). The authors use the fact that the XXX model with periodic conditions admits both presentations (inhomogeneous and homogeneous) to explore the former using the later.
As such the analysis presented in the paper is interesting, and can be viewed as a first step to deal with the inhomogeneous TQ relations in the general case.

The authors identify different types of Bethe roots for the ground state (real roots, pure imaginary ones that gather in strings and complex ones with arc-like structures). From their numerical studies (for system size 4 to 300), they conjecture the form of the ground state Bethe roots
in the thermodynamical limit and check the consistency of the conjecture from the comparison with the known result.

I think the paper is worth publishing, but some points deserve to be detailed in the conclusion.

The authors should indicate some directions on how to tackle the problem of the inhomogeneous TQ relations when there is no homogeneous TQ relations for the same model. Maybe the case of XXX model with twisted boundary conditions would be of some interest? Do they think that the structure observed for the Bethe roots in the periodic case is still valid for the twisted case?

From another point of view, a discussion on the possible analytical results that one could get is lacking. I am thinking in particular, to the recent results obtained
(in the lines of Yang&Yang's works, and many others) in arXiv:1508.05741 for the XXZ model with periodic boundary conditions: could a similar technics be applied for inhomogeneous TQ relations?

Once these points are detailed, I think the paper can be published.

Requested changes

Minor typos: 1- Bethe appears sometimes without upper case B

2- 2nd paragraph after eq. (4): the inhomogeneous TQ relation is referred as eq (2) instead of eq (3)

3- q(lambda) and Q(lambda) denote at the same time the Baxter operator(s) and their eigenvalues (maybe just put a hat on the operators, that you don't use so often)

---

## Round 2 · Referee Report · Junpeng Cao (Referee 2) · 2018-4-25

Strengths

1-, a interesting development of inhomogeneous Bethe ansatz equations

Weaknesses

1-, From Eq. (16), if Baxter T-Q relation is known, then this method works.

Report

The study of thermodynamic limit of the inhomogeneous Bethe ansatz equations is an important topic. In this manuscript, the authors reparameterized the Q-operator in the inhomogeneous T-Q relation of the periodic Heisenberg XXX spin chain by using the Lagrange polynomials representation. Then, they obtained the linear equations with respect to Q-operator and got the solution of Bethe roots of the inhomogeneous Bethe ansatz equations. Thus the distribution of Bethe roots at the ground state is studied. They found that the inhomogeneous term does not contribute to the Baxter T-Q equation in the thermodynamic limit. The different families of Bethe roots and their large N behavior are also conjectured and validated.

From Eq. (16), if Baxter T-Q relation is known, then this method works.

The results in this paper are reasonable and very interesting. I recommend this article for publication.

Requested changes

The necessary changes have been made by the authors in the new version [1803.09666v2]

---

## Round 3 · Referee Report · Anonymous · 2018-5-16

Report

The authors have answered to the points I raised, the paper can be published

---

## Round 3 · Referee Report · Junpeng Cao · 2018-5-17

Report

All my questions are responded and the manuscript is modified. I recommend this paper to be published.

---

## Round 3 · Author Response

We thanks the referee for is valuable comments.

---

## Round 3 · List of Changes

Following the Report 1, We add news comments in the conclusion and did the Requested changes.

---

## Editorial Decision

published